# Interpretable dimensionality reduction and classification of mass spectrometry imaging data in a visceral pain model via non-negative matrix factorization

Kasun Pathirage[1,2☯], Aman Virmani[1,2☯], Alison J. Scott[3], Richard J. Traub[4], Robert K. Ernst[3], Reza Ghodssi[1,2], Behtash Babadi[1,2]*, Pamela Ann Abshire[1,2]*

**1** Institute for Systems Research, University of Maryland, College Park, Maryland, United States of America, **2** Department of Electrical and Computer Engineering, University of Maryland, College Park, Maryland, United States of America, **3** Department of Microbial Pathogenesis, University of Maryland, Baltimore, Maryland, United States of America, **4** Department of Neural and Pain Sciences, University of Maryland School of Dentistry, Baltimore, Maryland, United States of America

☯ These authors contributed equally to this work.
* pabshire@umd.edu (PAA); behtash@umd.edu (BB)

**Data Availability Statement:** All MSI data files are available from the open science website Zenodo (doi: 10.5281/zenodo.7901681).

## Abstract

Mass spectrometry imaging (MSI) is a powerful scientific tool for understanding the spatial distribution of biochemical compounds in tissue structures. In this paper, we introduce three novel approaches in MSI data processing to perform the tasks of data augmentation, feature ranking, and image registration. We use these approaches in conjunction with non-negative matrix factorization (NMF) to resolve two of the biggest challenges in MSI data analysis, namely: 1) the large file sizes and associated computational resource requirements and 2) the complexity of interpreting the very high dimensional raw spectral data. There are many dimensionality reduction techniques that address the first challenge but do not necessarily result in readily interpretable features, leaving the second challenge unaddressed. We demonstrate that NMF is an effective dimensionality reduction algorithm that reduces the size of MSI datasets by three orders of magnitude with limited loss of information, yielding spatial and spectral components with meaningful correlation to tissue structure that may be used directly for subsequent data analysis without the need for additional clustering steps. This analysis is demonstrated on an MSI dataset from female Sprague-Dawley rats for an animal model of comorbid visceral pain hypersensitivity (CPH). We find that high-dimensional MSI data ($\sim$ 100,000 ions per pixel) can be reduced to 20 spectral NMF components with < 20% loss in reconstruction accuracy. The resulting spatial NMF components are reproducible and correlate well with H&E-stained tissue images. These components may also be used to generate images with enhanced specificity for different tissue types. Small patches of NMF data (i.e., 20 spatial NMF components over 20 × 20 pixels) provide an accuracy of $\sim$ 87% in classifying CPH vs naïve control subjects. This paper presents the novel data processing methodologies that were used to produce these results, encompassing novel data processing pipelines for data augmentation to support training for classification, ranking of features

**Funding:** This work was supported by a 2021 MPower Seed Grant from The University of Maryland Strategic Partnership: MPowering the State to authors PA, RE, RG, BB, AS, and RT. The funder website is https://mpower.maryland.edu/ The funders had no role in study design, data collection and analysis, decision to publish, or preparation of the manuscript.

**Competing interests:** The authors have declared that no competing interests exist.

according to their contribution to classification, and image registration to enhance tissue-specific imaging.

## Introduction

Mass spectrometry imaging (MSI) produces three-dimensional images in which each pixel at an $(x, y)$ location has a corresponding mass spectrum with mass-to-charge ($m/z$) and intensity axes. Raw MSI datasets can be difficult to interpret due to the sparse and distributed nature of the information, with many tissue characteristics associated with a combination of ions rather than individual ions. Analysis of replicate-powered MSI data presents significant challenges due to their large size, necessitating dimensionality reduction techniques and extraction of features for further analysis.

Most approaches to address dimensionality reduction in MSI do not preserve the inherent physical properties of MSI spectra, namely that MSI spectra are nonnegative. In this work, we explored the use of non-negative matrix factorization (NMF) as a dimensionality reduction technique that preserves spectral nonnegativity and provides a strong correlation with physiological features. Spectral peaks present in the extracted NMF components represent lipid ions present in the tissues.

This research establishes a new way to interpret MSI data that significantly reduces the data size and produces interpretable features, allowing for faster data processing and histological analysis based on MSI-derived features. We describe a data pipeline for extracting interpretable information from MSI data that can be represented compactly while preserving the spectral and spatial interpretability of the compressed MSI data. We have applied our data pipeline to two important applications- biological classification and generation of tissue histology images- to study the viability of our methods.

This paper presents novel approaches for MSI data analysis that build upon existing methods in three distinct ways: 1) by introducing a data augmentation technique that allows the use of NMF components for classification into biological groups using limited training data; 2) by introducing a statistical approach that may be used to extract biologically relevant, class-distinctive latent variables and to rank their contributions to the classification accuracy; and 3) by introducing an image registration technique that enhances the tissue-type specificity and correlation with H&E-stained images. The approaches are demonstrated on an MSI dataset for a rodent model of chronic visceral pain.

## Background and related research

This paper builds on existing techniques in mass spectrometry imaging, H&E staining, data compression, and data classification, which are briefly summarized below.

### Mass spectrometry imaging

Mass spectrometry imaging (MSI) is an analysis technique that generates a spatial distribution of ions and abundances in a given sample and can be used for a variety of molecular targets. Several MSI processes use different ionization techniques, with the most widely used being matrix-assisted laser desorption ionization (MALDI), desorption electro-spray ionization (DESI), and secondary ion mass spectrometry (SIMS), and their uses are well-reviewed [1–3].

The spatial aspect of MSI makes it possible to obtain anatomical images of any ion detected in the mass spectra in a given experiment. MSI has been widely used to map diverse analytes,

but it is particularly effective in analyzing lipids [4, 5] and has been used to map lipids and lipid fine structure in brain tissue [6], simultaneously map host and bacterial lipids [7], liposomal drug distribution [8, 9], and cancer [10] in tissues.

## H&E staining

H&E staining is a two-dye staining technique that is commonly used to evaluate tissues [11, 12]. It is also used in tandem with mass spectrometry [13]. The differential properties of the two dyes, hematoxylin and eosin, enhance the contrast of tissue features when observed under a microscope. Hematoxylin stains genetic material a blue-purple color, highlighting structures such as ribosomes and chromatin within the nucleus. Eosin stains cytoplasmic structures, highlighting cytoplasm, cell wall, collagen, and connective tissue in varying shades of pink [14]. H&E staining helps to discriminate between different types of cells and tissues and provides an important tool to understand the patterns, shapes, and arrangement of cells in a tissue sample [15, 16]. However, the evaluation of H&E-stained tissue still relies on the expertise of a trained pathologist or histologist; this process can be tedious and time-consuming, and there are abundant examples of similar pathologies that are not well resolved using H&E alone. The development of automated image segmentation to rapidly isolate regions of interest from standard stainings is an active area of interest.

## Data compression

MSI datasets can be very large ($\sim$ GB) depending on factors including spectral range, spatial sampling, and density of spectral data collection, with the potential for millions of ions to be represented at each location in a tissue sample. Dimensionality reduction techniques simplify the analysis of such datasets by representing MSI data compactly with minimal loss in information. Verbeeck *et al.* describe several unsupervised machine-learning approaches for MSI data analysis [17]. They compare principal components analysis (PCA) and NMF as dimensionality reduction techniques for MSI data, assessing the interpretability of extracted features using a synthetic dataset with known composition. They further report the ability of NMF to extract anatomically relevant regions in brain tissue imaged with MALDI MSI. Nijs *et al.* compared several dimensionality reduction algorithms including NMF, PLSA, LDA, and KL NMF, and found that NMF provides the best fit overall for MSI data [18]. Paine *et al.* were able to identify different compounds in cancer tissue from NMF spectra [19], establishing that NMF yields meaningful spectral components with peaks attributable to compounds present in the sample. Another important characteristic of the spatial components produced by NMF is its strong spatial correlation with anatomical tissue structure, which enables its capability to produce segmented views of tissue features. Trindade *et al.* used the spatial distributions of NMF components to differentiate similar but distinct resin types [20].

## MSI data processing

Several authors have reported data processing methods to extract and interpret information from MSI data. These methods involve clustering of spectral and spatial features to extract tissue characteristics as well as annotation of metabolites of interest. Many clustering methods exist to interpret the spatial information in MSI data such as k-means, GMM, and TSNE. Clustering on MSI data is computationally expensive and is usually preceded by dimensionality reduction. Prasad *et al.* evaluated several clustering methods using both real and synthetic MSI data and found that clustering performance decreased with increasing complexity, and data compression prior to clustering improves the performance [21]. In our analysis based on novel visceral pain data, we found that performing NMF across multiple tissue samples inherently

produced meaningful spatial distinction of the components without the need for explicit clustering.

Recent data processing approaches have reported new ways to incorporate either additional spectral or spatial features into MSI data after dimensionality reduction to extract interpretable information. Smets *et al.* incorporated spectral information in addition to spatial information by adding prioritization of selected *m/z* values to uniform manifold approximation and projection (UMAP) spatial embeddings [22]. Smets *et al.* have also reported an approach to combine molecular data from multiple UMAP spatial embeddings with histology data by creating low-dimensional 3D representations of RGB images which are fused using an adjustable parameter based on H&E data [23]. We have observed that NMF-compressed data inherently produces meaningful and interpretable features reflecting histology as well as spectral information and in this work have explored spectral and spatial representations based on linear combinations of NMF features.

Zhang *et al.* report a method that uses patches for data augmentation for training an ML model for subsequent dimensionality reduction and clustering [24]. The method presented in our paper also uses patches for data augmentation—but in this case for training of a classifier. The distinction is that the patches in our paper have already passed through a dimensionality reduction algorithm (i.e., NMF) whereas the Zhang *et al.* patches are taken directly from the raw MSI data and are used to train a dimensionality reduction algorithm. Unlike in this work, the Zhang *et al.* methodology allows spatially overlapping patches, which is suitable for training a dimensionality reduction algorithm but would introduce bias into the training of a classifier.

## SVM classification

Support vector machines (SVM) are a class of supervised machine learning algorithms that are mostly used for classification and regression problems [25]. SVM is widely used in the data analysis of biological and other sciences [26]. SVM operates by finding a decision boundary with the maximum margin, i.e., one that is farthest away from all classes. The decision boundary in general can be quantified over a higher dimensional space than the ambient space of the features, giving rise to Kernel SVM, in which the kernel defines the high-dimensional feature mapping. Examples of such kernels are the Radial Basis Functions (RBF) and polynomial kernels [25]. When the kernel is the identity mapping, the resulting SVM is known as linear SVM. For linear SVM, the classifier is equivalent to a linear combination of the features that is discriminative of the classes. While Kernel SVM typically achieves higher classification accuracy [25], it results in more complex and often less interpretable models. Linear SVM, on the other hand, generates simpler models whose weights may be used to identify the latent features that contribute to the classifier's performance [27].

## Methodology

### Ethics statement

This study was carried out in strict accordance with the recommendations in the guide for the care and use of laboratory animals of the National Institutes of Health and the guide for the use of laboratory animals by the International Association for the Study of Pain. The protocol was approved by the Institutional Animal Care and Use Committee (IACUC) at the University of Maryland, Baltimore (Protocol Number: 0220020).

## Animal model

Nociplastic pain describes chronic pain conditions that are not due to injury or disease (e.g., temporomandibular disorder (TMD), irritable bowel syndrome (IBS), fibromyalgia, migraine headache). Human patients often experience two or more conditions resulting in comorbid or chronic overlapping pain conditions (COPCs). Stress modulates colonic pain through activation of the hypothalamic-pituitary-adrenal (HPA) axis and the sympathoadrenal medullary (SAM) axis evoking the release of inflammatory mediators sensitizing colonic afferents. This leads to the hypothesis that the transition from normal sensory processing in the GI tract to chronic visceral pain involves changes in metabolic processing in the colon. In animals, orofacial inflammation followed by stress results in chronic visceral hypersensitivity modeling pain in patients with TMD and IBS [28, 29]. Using this comorbid pain hypersensitivity (CPH) model in female rats, colon tissue was collected at a period of heightened visceral hypersensitivity.

Female rats (Envigo; 10 weeks old at arrival at University of Maryland, Baltimore, animal facility) were acclimated to the animal facility for one week. Naïve rats (n = 4) were left in their home cage under normal husbandry conditions for 3 weeks. Rats were then euthanized by $CO_2$ asphyxiation followed by decapitation and tissue harvest. Following one week of acclimation, CPH rats (n = 4) were briefly sedated with isoflurane, and Complete Freund's Adjuvant (CFA; Sigma-Aldrich, F5881; 50 μL, 1:1 in saline) was injected into both masseter muscles. Starting the following day restraint stress was produced by placing rats in Broome-style rodent restrainers (4.8 cm diameter, 20 cm length) preventing movement for 2 hrs per day for 4 consecutive days. Rats were tilted at a 45-degree angle head up or head down in 15-minute blocks alternating with 15-minute blocks in the horizontal position. Two weeks after the last stress session, rats were subject to colorectal distention (3 trials of 20, 40, 60 mmHg distention, 20 sec each, 3 min interstimulus interval). Rats were subsequently euthanized by $CO_2$ asphyxiation followed by decapitation and tissue harvest.

## Tissue preparation

Colons were collected from naïve and CPH groups (n = 4 ea.) from cecum to anus and placed into a petri dish with room temperature porcine gelatin in endotoxin-free water (2% w/v; Sigma G1890). Gelatin solution was injected (∼200 μL) at five evenly distributed points along the colon length using a 21G needle. The colons were split along the mesenteric line and fecal material was removed. The anal junction was grasped with two flat wooden toothpicks and rolled, lumenal side inward, toward the cecal junction. The colon rolls were then placed upright on a foil boat, float-frozen on a pool of liquid nitrogen, sealed and stored at –80˚C before sectioning. Colon tissues were removed, prepared, and frozen in less than five minutes. Serial cryosections were collected on a Leica CM1950 (Leica Biosystems) starting from at least 1/3 of the cross-sectional depth of the rolled tissue at 12 μm thickness and thaw-mounted on indium tin oxide (ITO) glass microscope slides (Delta Technologies). This preparation orients the proximal colon on the outer rings of the roll and the distal colon in the center. Slides were stored at –80˚C prior to data collection. At the time of analysis, glass slides were placed in a vacuum desiccator to thaw (less than five minutes total). An orientation light scan was collected on a flatbed scanner.

## Mass spectrometry imaging and staining

Sections were coated with norharmane (NRM) matrix solution of 7 mg/mL in 2:1 (v:v) chloroform:methanol using an HTX M5 Matrix Sprayer (HTX Technologies, NC). The following matrix application settings were used: 10 passes, 10 psi, 2 L/min nitrogen gas, 30C nozzle, 40

mm height, 0.1 mL/min, 1200 mm/min velocity, CC pattern, and 2.5 mm track spacing. Data were collected on a Bruker timsTOF flex (Bruker Daltonics) instrument in negative ion mode from $m/z$ 600–2000. The instrument was calibrated to the Agilent ESI peptide standard mix resulting in a sub-ppm standard deviation calibration. The MALDI laser was operated in the M5 small setting with 16 $\mu$m x 16 $\mu$m beam scan resulting in 50 $\mu$m spatial resolution. Data used in this work were collected using MALDI negative mode MSI since it has a wide range of molecular masses, and negative ion mode lipid data provides excellent reproduction of the details of tissue structures [30].

Raw data were individually imported into SCiLS Lab software [31] as centroided data on loading and the individual files were exported to the common data format, imzML [32] for further analysis. Following MSI, the matrix was cleared with two consecutive dips (10 seconds each) in 70% ethanol and tissue stained with H&E as previously described [30]. Slides were cleared in xylene and permount was used to attach coverslips. Optical images were collected on an Aperio slide scanner (Leica Biosystems) at 20x magnification and images were exported in eps format from Leica's ScanScope software.

## Datasets

8 MSI datasets along with their respective H&E-stained images were generated using tissue samples from the 4 CPH and 4 naïve animals. The 8 datasets collectively will be referred to as the 'data cohort', while the term 'dataset' will refer to the MSI data corresponding to a single tissue sample. Fig 1 shows ion images from the 8 datasets at $m/z$ 885.5, with samples from CPH animals shown on the left, and naïve control animals shown on the right. The chosen ion at $m/z$ 885.5 is suspected to be the lipid ion 1-stearoyl, 2-arachidonyl-phoshphatidylinositol (SAPI), which is known to play a role in pain hypersensitivity in animals [7]; however, the identity of this lipid candidate has not been confirmed.

## Data processing pipeline

The Python programming language running on a Dell Precision 5820 (Intel Core i9 10900X CPU with 20 cores) with 256 GB system memory was used to process and analyze the data. The Python library pyimzML [33] was used to parse the data from imzML format to the computer memory. Fig 2 summarizes the data processing pipeline which is explained in the steps below.

- **Step 1: Data binning** This experiment resulted in large individual datasets ($\sim$ 13 GB per dataset) and is therefore saved in a sparse file format. Binning, as the pre-processing step,

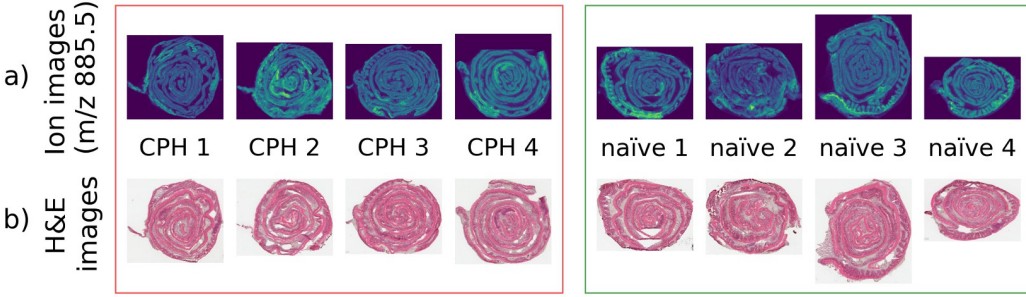

**Fig 1. The biological replicates.** (a) Mass-spectrum image at $m/z$ 885.5, and (b) the H&E-stained image for each biological replicate. The CPH and datasets are shown as groups on the left and right respectively.

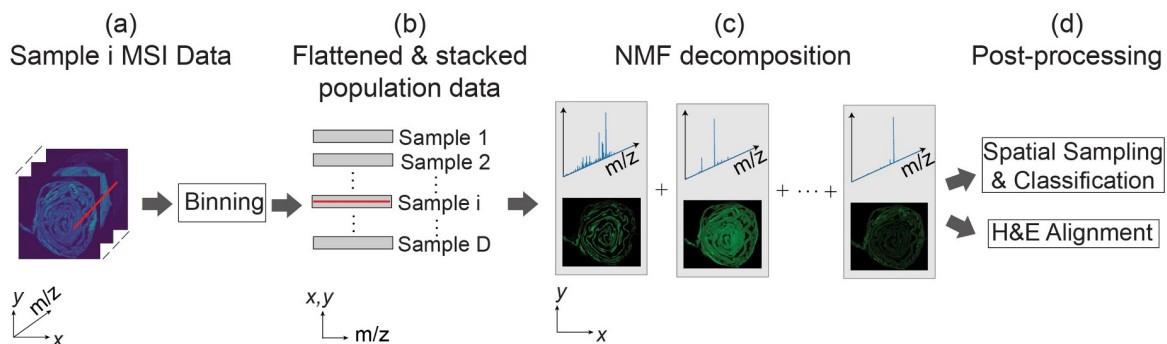

**Fig 2. Overview of the data pipeline.** (a) Raw data is binned, truncated, and normalized. (b) The 3D image-spectral datasets are flattened and stacked into 2D matrix form. (c) Computation of NMF corresponding to spatial-spectral decomposition. (d) NMF features are used for further processing including classification and histological analysis.

offers a two-fold advantage. First, it allows us to down-sample the data to a lower resolution to make the computations much faster in the subsequent steps. Second, it enables matrix calculations that are needed later, by transforming the data into uniform and equally spaced *m/z* bins. This is shown in Fig 3.

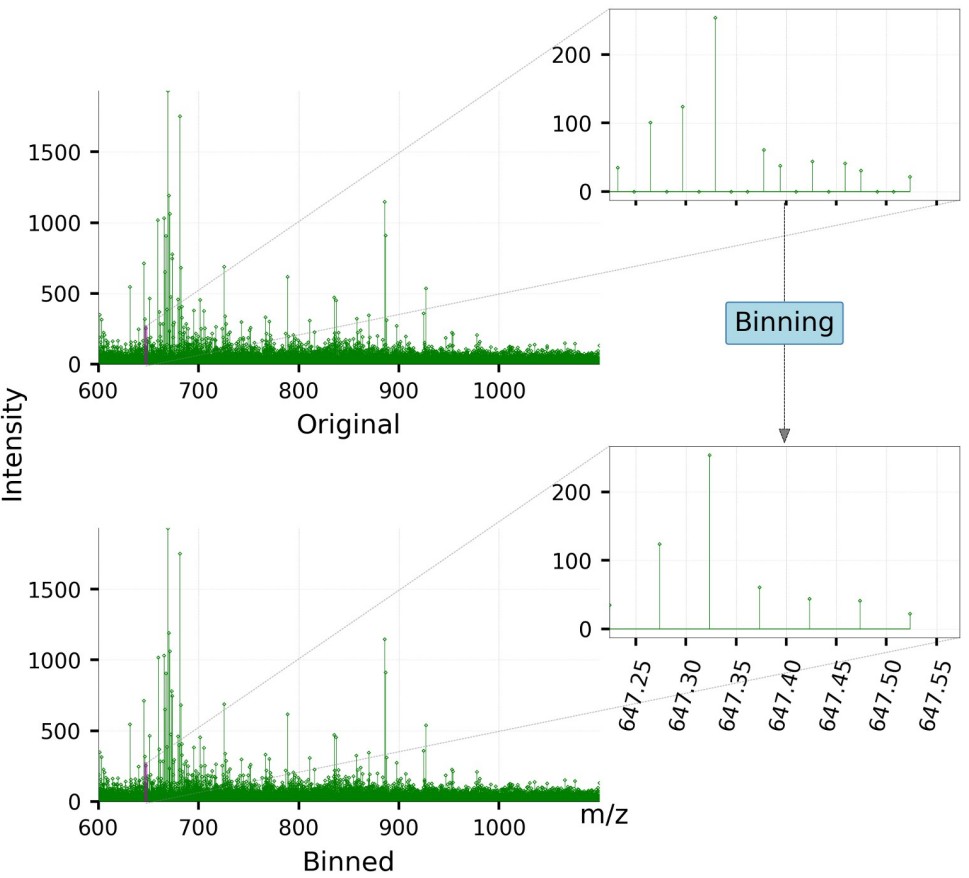

**Fig 3. MSI spectra before, and after binning.** Original MSI spectra are binned to bins of width 0.05 *m/z* to create uniformly spaced peaks. The bin size of 0.05 *m/z* sufficiently preserves the spectral resolution of the MSI data as can be seen through the insets.

The sparse nature of MSI datasets allows for spectral binning with minimal information loss. We used a bin width of 0.05 Da and maximum peak intensity (ion abundance) within a bin to represent the bin intensity. Binning at the 0.05 Da bin width reduces the spectral dimension from 100,000 to 28,000 features.

The binned data was stored in memory in a 3-dimensional array of size $(A^d \times B^d \times M)$ per dataset. Here, $A^d$ and $B^d$ are the number of pixels in the horizontal and vertical dimensions for a given dataset $d$ which form an 'ion image' for each detected $m/z$ value. A dataset can therefore be understood as a stack of $M$ images, each with a size of $A^d \times B^d$ pixels. Binning leads to $M$ being consistent for every pixel $p$ in every dataset in the cohort. It should however be noted that the $A^d$ and $B^d$ values are different for each dataset $d$. This is because the tissue samples from different animals may take different physical shapes and sizes.

- **Step 2: Truncation** Although each spectrum ranged from 600 Da to 2000 Da, we found that the spectra became much sparser beyond 1100 Da. This corresponds to the upper limit of the typical phospholipid mass range and was subsequently truncated to 1100 Da. This truncation further reduced the spectral dimension from 28000 to 10000 features.

- **Step 3: Normalization** The binned data is subsequently normalized based on the total ion current (TIC) measure using the formula in Eq 1.

$$\tilde{I}^d_{x,y,s} = \frac{I^d_{x,y,s}}{\sum_{s=1}^{M} I^d_{x,y,s}} \tag{1}$$

Here $I^d_{x,y,s}$ and $\tilde{I}^d_{x,y,s}$ are the raw and TIC normalized intensities of the $s^{th}$ bin center ($m/z$ value) at the $(x, y)^{th}$ pixel location of the $d^{th}$ dataset respectively, and $M$ is the total number of bins in each pixel location $(x, y)$.

TIC normalization is an essential part of the pipeline. The sum of intensity values $\sum_{s=1}^{M} I^d_{x,y,s}$ for a TIC normalized spectrum at pixel location $(x, y)$ adds up to 1.

- **Step 4: Dimensionality reduction** After preprocessing the datasets as described in steps 1–3, we perform dimensionality reduction separately with NMF and with PCA. While the following steps describe the steps used with NMF, many of the same considerations apply to PCA.

  1. **Step 4a: Flattening and stacking each dataset** We first flatten the 3-dimensional datasets of size $(A^d \times B^d \times M)$ into 2D datasets of size $(A^d B^d \times M)$, and stack them along the combined $(x \cdot y)$ spatial axis for input to the NMF algorithm. This ensures that NMF finds basis vectors that are common to all datasets. Since NMF does not change the data order along the rows, we are able to separate the low-dimensional output corresponding to each dataset from the stacked output.

  2. **Step 4b: Computing the NMF spatial and spectral features** Stacking of the datasets leads to a combined data array **I** of dimensions $(\tilde{N} \times M)$, where $\tilde{N} = \sum_{d=1}^{8} A^d \times B^d$ is the total number of pixels in all datasets in the cohort. We performed NMF on this combined data array, reducing the dimensionality $M$ from 10,000 to 20. This reduced number of dimensions is denoted by $m$.

     The NMF algorithm compresses raw MSI data as given in Eq 2.

$$\tilde{I}^d_{x,y,s} = \sum_{j=1}^{m} Z^d_{x,y,j} \cdot \Psi_{s,j} + E^d_{x,y,s} \tag{2}$$

where the reduced dimension representation for $\tilde{I}_{x,y,s}^{d}$ is defined at spectral bin $s$ $(m/z)$ and 2D spatial location $(x, y)$ for the $d^{th}$ tissue sample (with $d = 1, 2, \ldots .D$), $Z_{x,y,j}^{d}$ is the $j^{th}$ *spatial* NMF component at location $(x, y)$ and $\Psi_{s,j}$ is the $j^{th}$ *spectral* NMF component at $m/z$ bin $s$, and finally $E_{x,y,s}^{d}$ is the residual error that cannot be captured by the NMF decomposition. Note that the spectral NMF component is sample-independent to account for the population-level spectral composition, whereas the spatial component is sample-dependent to account for the sample-specific spatial variations. The NMF components are estimated by minimizing

$$\sum_{d=1}^{D}\sum_{x,y,s}\left|\tilde{I}_{x,y,s}^{d} - \sum_{j=1}^{m} Z_{x,y,j}^{d} \cdot \Psi_{s,j}\right|^{2} \tag{3}$$

subject to the non-negativity constraints $Z_{x,y,j}^{d} \geq 0, \forall x, y, j$ and $\Psi_{s,j} \geq 0, \forall s, j$ [34 35], in which the optimization problem in Eq 3 is typically solved using iterative methods.

This reduced feature space contains 20 basis vectors, (spectral NMF components; $\Psi$), and 20 low-dimensional features (spatial NMF components; $\mathbf{Z}$). The portion of each spatial NMF component ($Z_{\cdot,\cdot,j}^{d}$) corresponding to each dataset $d$ can be reshaped into an image describing the spatial distribution of lipid ions contained in its corresponding spectral component. The output of the NMF algorithm is the transformed MSI data with 20 *component spectra* and 20 *spatial intensity maps*.

The number of NMF components was selected to be 20 based on the normalized residual reconstruction error as shown in Fig 4. The normalized reconstruction error is defined as the sum of the squared difference between the binned MSI data and its NMF reconstruction, normalized by the sum of squares of the binned MSI data. It falls quickly for the first few NMF

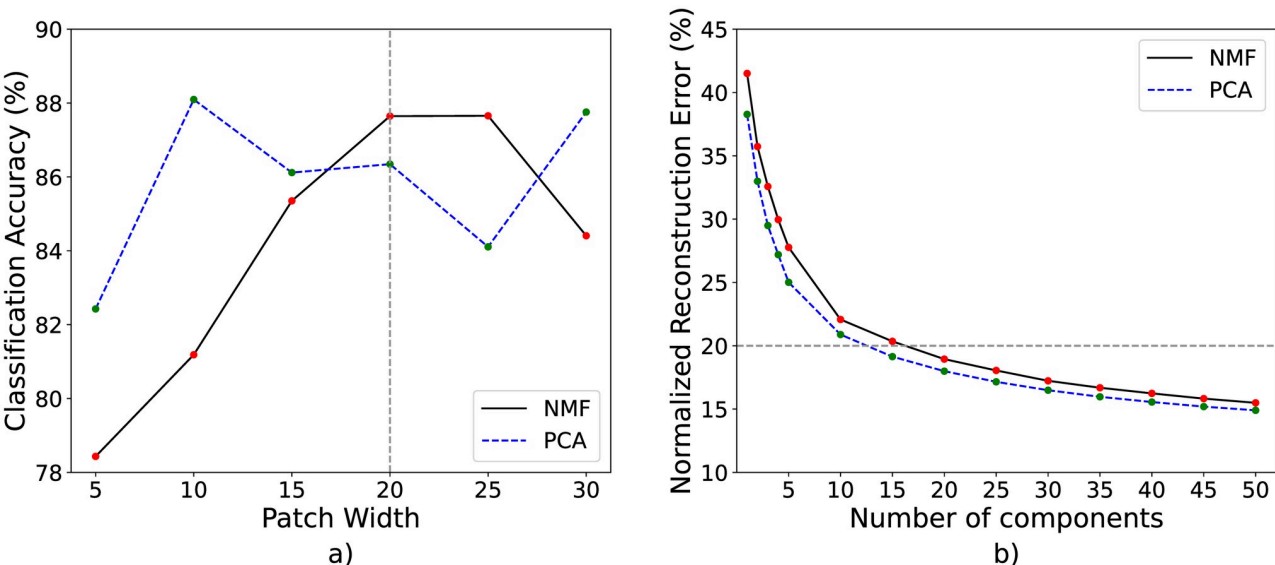

**Fig 4. Performance for NMF and PCA data compression in classification and data reconstruction.** (a) The relationship between SVM classification accuracy and the width of patches extracted from the NMF spatial intensity maps. The gray dotted line shows the patch width of 20 × 20 pixels used to generate the results presented in this paper. (b) The relationship between reconstruction error (normalized root-mean-squared error as a percentage) and the number of PCA/NMF components. The gray dotted line marks the normalized error of 20%.

components and then more gradually, with the reconstruction error falling under 20% for 20 NMF components. Results in the remainder of the paper are presented for 20 NMF components.

We used the NMF algorithm available with the scikit-learn package for Python [36]. 6000 iterations with the default parameters were required for convergence. The sklearn NMF library does not sort NMF components according to any measure. Therefore, we use a backward elimination technique to rank them based on their contribution to the reconstruction of the binned MSI data. Starting with the full set of 20 NMF components, we remove one component and calculate the residual reconstruction error between the binned MSI data and its reconstruction using the remaining NMF components. The component that leads to the highest residual reconstruction error when removed is ranked as the most important component for the reconstruction task and is designated as component 0. This process is repeated until only 1 NMF component is left, which is the least important component for reconstruction and is designated as component 19.

## Classification pipeline

This subsection describes the methodology used to train a support vector machine (SVM) classifier to distinguish between CPH and naïve data. High classification accuracy is one of the *necessary* conditions for the presence of 'pain-related metabolites' in CPH animals, and the absence of such in naïve animals. However, it should be noted that this is not a *sufficient* condition for the hypothesis to be considered true. The steps in the pipeline are described below.

- **Step 1: Spatial reconstruction** In the dimensionality-reduction step, all the datasets were concatenated into a single array. To perform the classification, the datasets need to be separated and labeled as being from a CPH dataset or a naïve one. After separating and labeling the data, the NMF spatial intensity map for each dataset $d$ was reconstructed by reshaping the data into a 3-dimensional array of size $(A^d \times B^d \times m)$. It is important to notice the resemblance this has to the initial binned, truncated, and normalized 3-dimensional data array $\tilde{I}^d_{x,y,s}$ mentioned in the data processing pipeline above. The key difference is that the depth dimension has now been reduced from 10,000 to 20. This modified data cohort will henceforth be referred to as the 'compressed data cohort'.

- **Step 2: Data augmentation** The 8 datasets in the compressed data cohort in their raw form would only contribute 8 labeled data samples for the classification task. Data augmentation was therefore required to prevent overfitting of the SVM classifier. As shown in Fig 5, augmentation was achieved by redefining a data sample as a spatially cropped version of a dataset from the compressed data cohort. Each dataset was spatially divided into a grid of non-overlapping patches with each patch being of size $(20 \times 20)$ pixels. All patches that were off-tissue were discarded from the augmented dataset. This defines a data sample to be a 3-dimensional array of size $(20 \times 20 \times m)$, where $m$ is the number of components in the feature space after dimensionality reduction and takes a value of 20. This approach makes the implicit assumption that the metabolites leading to pain in CPH animals are distributed throughout the entire tissue area.
The 3-dimensional data samples were flattened into 1-dimensional vectors of size $(8000 \times 1)$ in preparation to be fed into an SVM classifier. The data augmentation step generated a total of 2,000 labeled sample vectors.

- **Step 3: Data split** 80% of the data patches were used to train the classifier. The remaining 20% were used as a testing set. Both the training and testing sets were balanced such that there were equal numbers of samples for both the classes CPH and naïve.

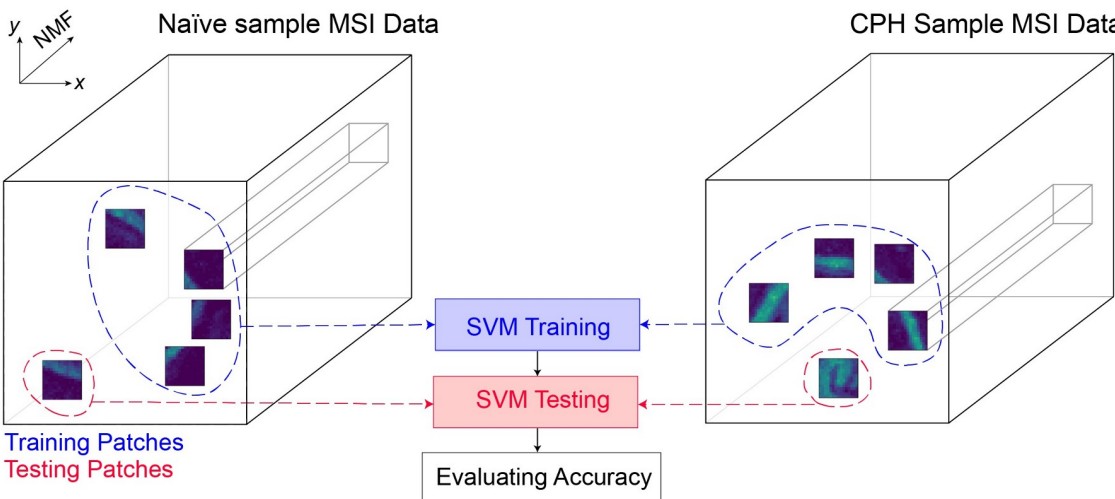

**Fig 5. SVM classification and data augmentation methodology.** 20×20×20 patches of NMF data sampled from different regions of labeled CPH and naïve datasets were used to train and test the SVM classifier.

- **Step 4: SVM classification** Binary classification was performed with the positive class corresponding to CPH data and the negative class corresponding to naïve data. A high classification accuracy would therefore establish a *necessary* (but not *sufficient*) condition to infer that the CPH animals had certain features in their MSI data that correlated with their hypersensitivity to pain. We used the Support Vector Classifier (SVC) module of the Python sklearn library [36], evaluating both linear and radial basis functions (RBF) as kernels. We used 5-fold cross-validation, implemented with the GridSearchCV module of the Python sklearn library to tune the hyperparameters of the SVM algorithm.

## Ranking NMF components according to their contribution to classification accuracy

One of the goals of this study is to test the hypothesis that there exists certain lipid ion features or a cluster of features in CPH animals that correlate with sensitivity to pain. This hypothesis is tested with the results of the SVM classification. We identified the latent variables that support this hypothesis by evaluating which NMF components were most important in discriminating between CPH and naïve animals. We call these NMF components "the candidate list" as their spectra may contain lipid ions that are associated with pain hypersensitivity.

The data cohort used in this study contains images from 8 different animals with natural differences in the size of the tissue segments, resulting in different spatial sizes for the MSI ion images as well as the NMF spatial intensity maps. Images of different sizes produce different numbers of patches, and including all patches will bias the classification toward animals with larger MSI ion images. To overcome this bias, we developed a statistical sampling methodology, as follows:

1. Determine the dataset with the smallest number of patches, *P*.

2. Randomly select *P* image patches from each dataset.

3. Use 5-fold cross-validation to train and test $N$ SVM classifiers, each using only the intensity map corresponding to a *single* NMF component. This results in $N$ values for mean cross-validation classification accuracy.

4. Return to step 2, selecting a different random set of patches for each of the non-minimum size images.

5. Repeat 50 times to obtain a distribution of classification accuracies for each of the $N$ components.

To rank the 20 NMF components, we use an iterative approach. During the first iteration, $N$ takes the value 20; i.e., the total number of NMF components. We perform two-sample unpaired t-tests to evaluate the statistical significance of the mean of each distribution against that of the distribution that has the highest mean [37]. We select the component with the highest mean and any other components that have mean accuracies statistically similar to it and append them to the candidate list. Let us now assume that the candidate list has $K$ candidate components at the end of this first iteration. During the next iteration, we repeat steps 1–4 above on the candidate list, augmented by single NMF components that are not yet selected (i.e., $N$ takes a value of $20 - K$ now) This process is repeated until the stopping criterion is reached, which in this case was to expand the candidate list until the components in the candidate list alone can produce a classification accuracy of at least 75% (i.e., 25% above chance level). This procedure generates a list of components that are ranked according to their impact on classification accuracy. This process is computationally intensive, so we used the University of Maryland supercomputing cluster to accelerate the computations through parallelization of the repeat calculations.

## Image registration pipeline

Generating biological and mechanistic insight from untargeted spatial 'omic information in raw MSI data is generally a difficult task due in part to its high dimensional nature. Further, tissue annotation is a time-consuming step that requires expert-level evaluation for complex pathology. In MSI, each ion will have its corresponding ion image. However, not all ions will have a distinctive spatial structure. In contrast, an H&E-stained image contains a multitude of anatomical features with fine spatial resolution, but without well-defined mapping to different tissue types due to the limited specificity available from shades of the two stains applied to the tissue.

Although H&E staining and MSI are separate modalities carrying separate types of information, their complementary nature enables augmenting the anatomical features visible in the H&E-stained images with information from the NMF spatial intensity maps. However, MS images and H&E-stained images have differences in scaling and non-linear perspective distortions as they are acquired using disparate types of instruments. Therefore, to effectively compare the NMF spatial intensity maps generated from MSI with H&E-stained images, we follow the image registration pipeline described below to align these images.

**Step 1: Scaling** Image alignment algorithms typically require the input images to be scaled to the same size. Therefore, the H&E-stained images which have higher *pixel* resolution are down-sampled to match the size of their corresponding NMF spatial intensity maps. This was accomplished using the Python OpenCV library [38].

**Step 2: H&E-stained image segmentation** The scaled H&E-stained images are decomposed into segments based on the color profile of visible spatial structures such as the muscular lining, mucosal layers, and regions of immune cell aggregation. As an example, to segment the muscular lining, a region of interest (ROI) $8 \times 8$ pixels in size is defined on top of the

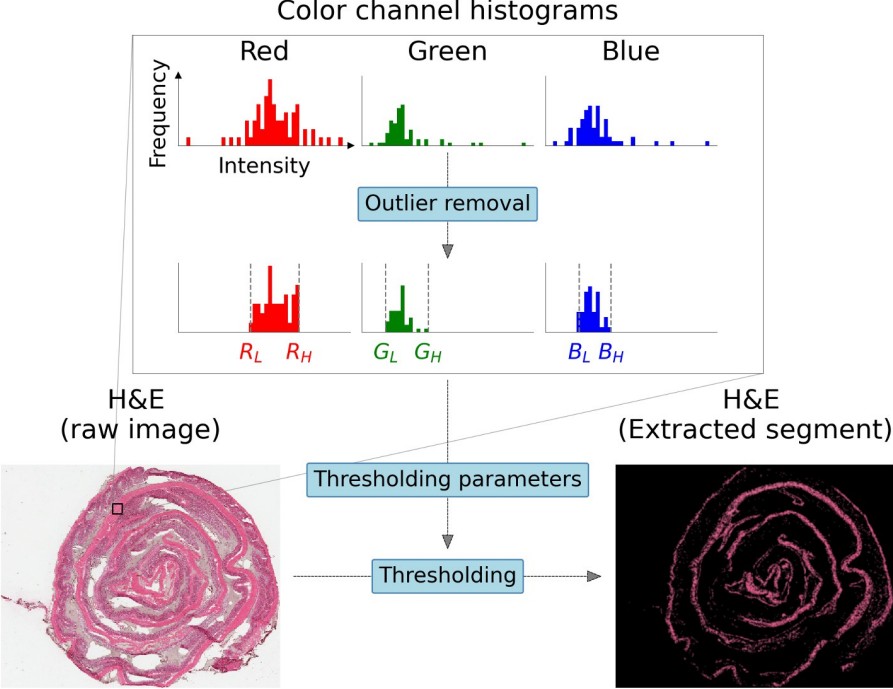

**Fig 6. Simplified methodology for anatomical feature segmentation of H&E-stained images.** Red, green, and blue color channel intensity distributions are extracted in a region of interest (ROI) centered within the boundaries of the desired anatomical structure. Following outlier removal, 6 thresholds are determined based on the means and standard deviations of each color channel data as parameters to generate a mask that can be applied to the original H&E-stained image to segment the desired anatomical feature.

muscular lining in an H&E-stained image. The means $\mu^c$ and standard deviations $\sigma^c$ of each color channel $c$ (red, green, blue) within this ROI are calculated. The entire H&E-stained image is subsequently thresholded to extract pixels with intensities within two standard deviations of the mean ($\mu^c \pm 2\sigma^c$) of each color channel. Fig 6 shows the ROI determination and thresholding.

**Step 3: Edge detection** Edges act as landmarks that can enhance alignment. Therefore, we run the NMF spatial intensity maps and segments of H&E-stained images through a Canny edge detector available in the Python OpenCV library [39].

**Step 4: Homography transformation and alignment** We iteratively optimize a homography transformation algorithm in the Python OpenCV library between each H&E image segment and NMF spatial map. The optimal alignment between each image pair is obtained by maximizing the enhanced correlation coefficient (ECC) score [40]. The ECC score takes a high value if the NMF spatial map and the H&E image segment being aligned have similar spatial structures. For each dataset, we select the alignment that gives the highest ECC score and extract the corresponding warp matrix. We then apply the non-linear transformation defined by this warp matrix on the original H&E-stained image to obtain the desired alignment.

## Enhanced tissue-specific image generation

The NMF spatial intensity maps contain anatomically relevant features. Color-coding each spatial intensity map and overlaying them on top of each other generates a composite 'NMF-

based H&E-like image' that provides enhanced tissue specificity by highlighting different anatomical structures in different colors.

NMF spatial intensity maps contain pixels with both low and high intensities. Regions of low intensities generally correspond to noise or background and usually represent areas of a tissue that contain low abundances of ions defined by the respective NMF component's spectrum. If such low-intensity pixels are color-coded, they may overlap with information-rich high-intensity pixels of a different NMF component's spatial intensity map, thereby masking important information. Therefore it is important that the color-coding is only applied to pixels that have intensities above a certain threshold. This threshold may be tweaked depending on the context or depending on the contrast between the foreground and background pixels.

Once an appropriate threshold is determined, it is applied on the NMF spatial intensity maps to extract two binary masks; i) a *foreground mask* that defines the foreground pixel locations, and ii) a *background mask* that defines background pixel locations. We apply the *background mask* on the corresponding spatial intensity map to select and artificially set the below-threshold pixels to zeros. This way, only the foreground pixels will be color-coded during subsequent steps. To convert the grayscale NMF spatial intensity maps to their color-coded versions, we first determine an appropriate set of visually contrasting colors and obtain their {red, blue, green} vector mapping using a color vector table. A color vector is a three-element vector containing a value between 0 and 1 for each of the three color channels red, green, and blue. For example, the color vector (1, 0, 0) represents pure red while the color vector (0.1, 0.8, 0.3) represents a mixture of 10% red, 80% green, and 30% blue. The foreground pixels previously extracted are scaled by the color vector and assigned to three color channels to form a color-coded NMF spatial intensity map. This process is repeated for the remaining spatial intensity maps. As the final step, these color-coded NMF spatial intensity maps are combined into a single composite image by simply adding them together. To display images, they need to be converted to 8-bit unsigned integer arrays. The previous steps could generate certain pixels that have intensity values greater than 255, which cannot be represented by an 8-bit integer. Such pixels are artificially clamped at 255. It should be noted however that clamping of too many pixels could lead to a saturated image with poor contrast.

## Results

### Dimensionality reduction with NMF

Fig 7a shows the five NMF components that contributed the most towards reducing the reconstruction error, accounting for 70% of the reconstruction accuracy achieved with all 20 components. It should be noted that all spectral intensities are non-negative. The spatial components, i.e., the NMF spatial intensity maps, show diverse and distinct spatial structures. Data is shown from one representative dataset out of the 8 in the compressed data cohort.

### Dimensionality reduction with PCA

Fig 7b shows the five PCA components that explain the most variance in the data, accounting for 90% of the variance and 13% of the reconstruction accuracy achieved with 20 PCA components. It should be noted that certain spectral intensities are negative. Fig 7b also shows how the spatial information captured by the first few PCA components are feature-dense, but the presence of distinct spatial structure gradually tails off with an increasing number of components. In comparison, the NMF representation captures sparser spectral and spatial components with approximately equal numbers of spectral peaks and spatial features in each component.

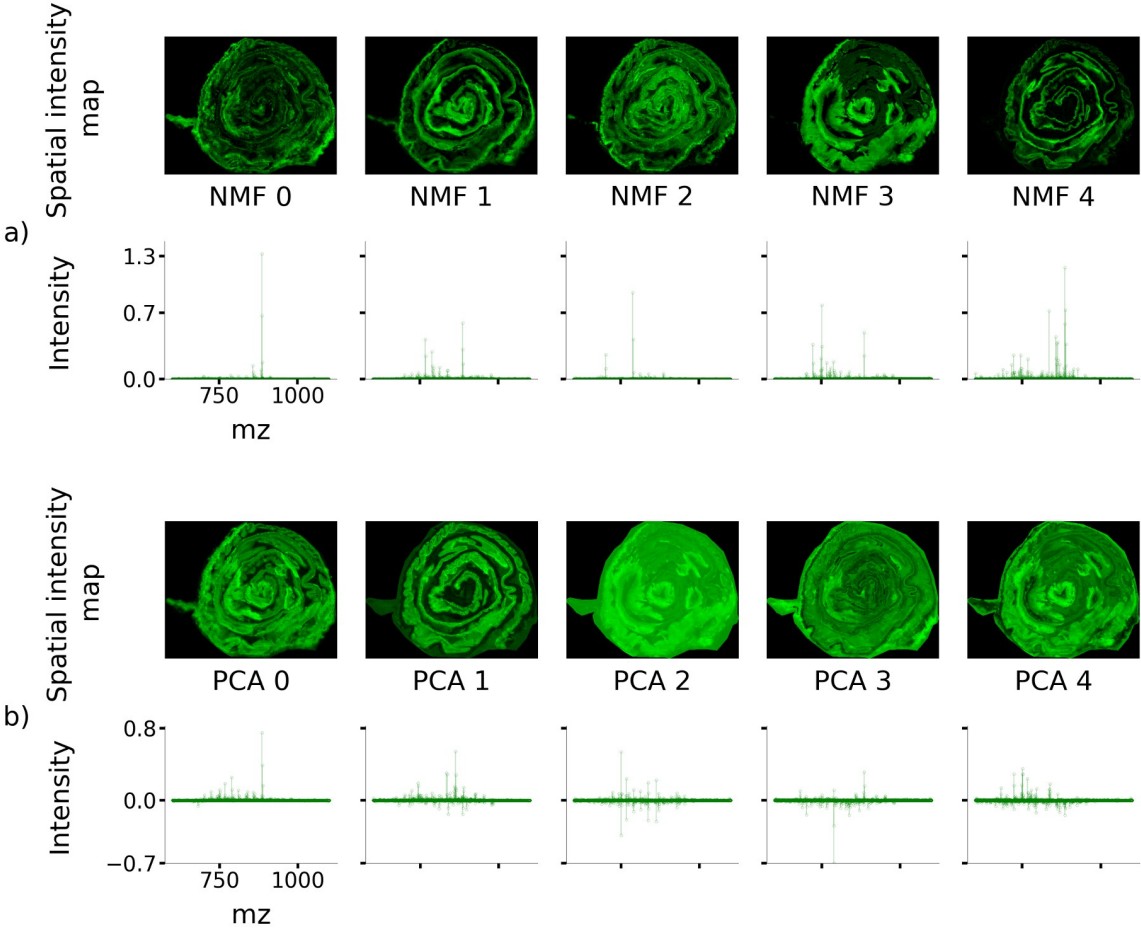

**Fig 7. The first five NMF and PCA components ranked according to their contribution towards MSI data reconstruction.** (a) First five NMF components. Observe how the NMF spectral peaks take only positive values. (b) First five PCA components. The PCA spatial maps gradually lose structure for higher-ordered PCA components.

## CPH vs naïve data discrimination

The $F_1$ score is a measure of classification accuracy reflecting precision and recall. The SVM classifier achieved $F_1$ scores of 99.9% and 87.5% on NMF training and testing data respectively. These results were achieved with a kernel SVM using the RBF kernel. SVM with a linear kernel achieved $F_1$ scores of 95% and 83% on NMF training and testing data respectively.

Fig 4 shows how the classification accuracy depends on the size of image patches used during the data augmentation stage. Note that the number of patches was maintained at a constant value during these analyses.

We found that the RBF kernel SVM classifier achieved a classification accuracy of 99.9% and 87.4% on PCA training and testing data respectively. Although the SVM classification accuracy is comparable for PCA and NMF data, NMF produces histologically meaningful spatial components, and directly interpretable spectral components compared to their PCA counterparts.

This study was carried out using an MSI data cohort based on single tissue samples from 8 rats split into two experimental groups, CPH and naïve. As shown in Fig 4, successful discrimination of samples from the two groups was obtained using an SVM classifier based on

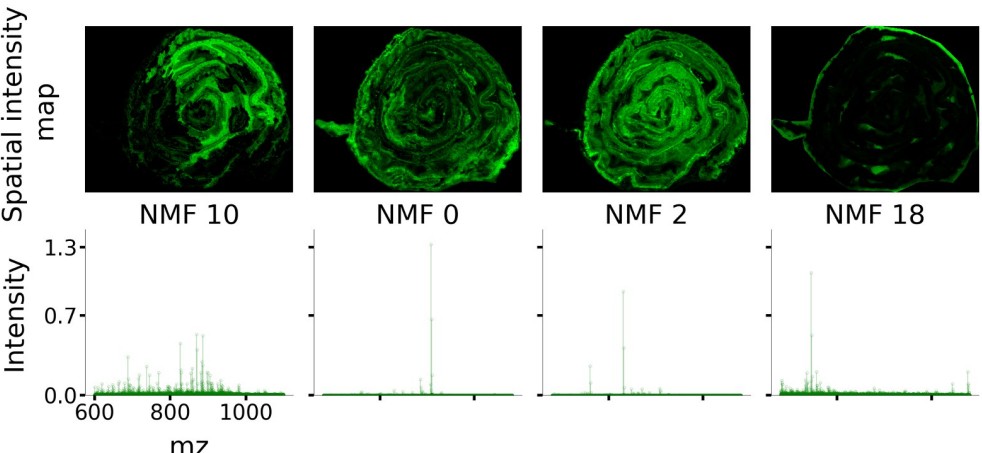

**Fig 8. Four NMF components that contributed the most towards discriminating CPH vs naïve data samples.** Note that the spatial maps are distributed throughout the colon structure indicating that pain-causing biomarkers may be found throughout the colon and fall into one of three classes: complex fingerprint, simple predominant ion, and off tissue.

relatively small ($20 \times 20 \times 20$) patches of NMF features. Classification accuracy generally increased with the number of NMF components used and with the patch size (saturating around 20 components). Classification accuracy increased from 75% for 5 NMF components, to 82% for 10 NMF components, to 87.2% for 20 NMF components (for $20 \times 20 \times 20$ patches). Classification accuracy increased from 78% for $5 \times 5 \times 20$ patches to a maximum of 87.65% for $25 \times 25 \times 20$ patches, decreasing for larger patches due to the limited number of samples.

## Most discriminating spectral features

Fig 8 shows the spatial and spectral distributions corresponding to the four NMF components that contributed the most toward classification accuracy. As explained in methods, these four components alone, when used in a *linear* SVM classifier yield a CPH vs naïve discriminatory $F_1$ score of 77.5% compared to 83% for all 20 components (this accuracy level should not be confused with the 87.5% $F_1$ score obtained when *RBF* kernel SVM was used with all 20 NMF components).

It can be observed in Fig 8 that the NMF components that contributed the most towards SVM accuracy are distributed throughout the swiss-roll structure and fall into three classes: NMF 10 exhibits a complex spectral "fingerprint" with moderate intensities, NMFs 0 and 2 reflect spectra predominated by high-intensity single lipid ions, and NMF 18 is an off-tissue component.

## Alignment of H&E-stained images with spatial components of NMF

Fig 9 shows a selected subset of the aligned NMF spatial intensity maps alongside H&E-stained images. It is notable that the NMF spatial components reflect spatially coherent regions of the tissue such as the muscular lining, submucosa, regions of inflammation, *etc.*, in the different components. As shown in Fig 10, by color-coding and overlaying individual NMF spatial intensity maps, an equivalent to an H&E-stained image with enhanced specificity for different tissue types can be obtained.

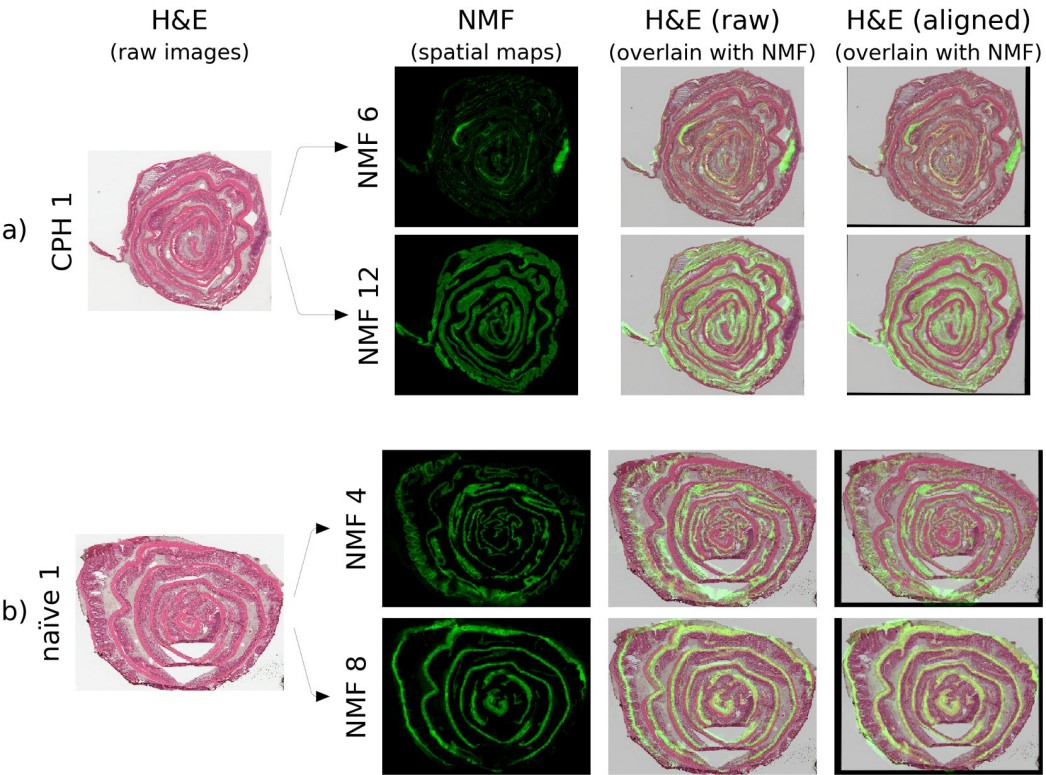

**Fig 9. Spatial features in NMF components align strikingly well with anatomical features in H&E-stained images.** a) Example alignment for a CPH dataset showing H&E image, original NMF spatial components, and the aligned images which enhance identification of tissue structure. b) Alignment for a naïve dataset. Similar features are apparent in the raw and H&E images. However, the correlation is moderate due to alignment mismatch. The aligned images demonstrate excellent correlation and preserve detailed tissue structure.

This indicates that NMF features can be used to identify tissue structures, at least in rat colon tissue. This may reflect underlying differences in the phospholipids present in the cell membranes for different tissues which the NMF spatial-spectral decomposition successfully retains. It is hypothesized that this observation may also translate to interpretable feature extraction in other types of tissue with complex histological features.

## Discussion

In this work, we discussed the high interpretability of *both* spatial and spectral components generated by NMF. The NMF spectral components are strictly non-negative, and can thus be interpreted to represent the presence of specific lipid ions corresponding to the mass-to-charge peaks in the spectra. We also found that NMF spatial intensity maps correlate strongly with spatial tissue structure and therefore could be used to obtain information typically captured with a different modality such as H&E-stained imaging.

### Comparison of NMF with PCA

We compared NMF feature extraction with PCA, a dimensionality reduction technique commonly used with MSI data. A comparison of quantitative and qualitative performance characteristics is shown in Table 1.

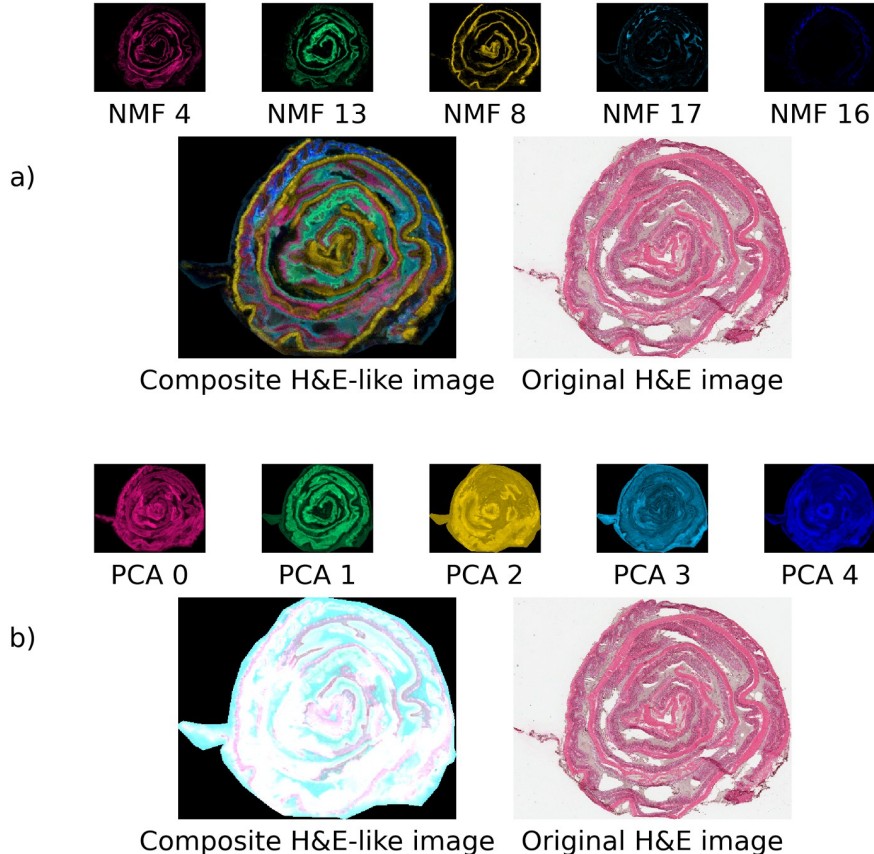

**Fig 10. Composite H&E-like images.** a) NMF spatial maps may be color-coded and combined to generate H&E-like images with enhanced contrast and spatial detail relative to actual H&E-stained images (despite the higher pixel resolution of H&E data). Only 5 components are represented for simplicity. The composite image contains many of the visible anatomical features of the original H&E image with greater specificity for different tissue structures. b) Color-coded PCA spatial maps do not overlay well to generate a high-contrast composite image. This is because only PCA 0 and PCA 1 have well-defined structures while the rest of the components are noisy, producing a smeared/saturated composite image.

The fact that PCA components are allowed to have negative values makes it difficult to interpret the meaning carried by the peak intensities in a PCA spectrum. A PCA component having large positive peak intensities may superimpose with another component carrying negative peak intensities to generate a reconstructed spectrum carrying zero intensity peaks.

**Table 1. Comparison of performance characteristics for PCA and NMF features on MSI data cohort from an animal model of comorbid visceral pain hypersensitivity.**

| Performance comparison | PCA | NMF |
|---|---|---|
| **Reconstruction error (20 components)** | 17.99% | 18.94% |
| **Classifier accuracy (20 components)** | 88.09% | 87.65% |
| **Spectral interpretability** | requires extra processing | nonnegativity allows molecular ID |
| **Spatial interpretability** | 1–2 components correlate with histological features | most components exhibit significant correlation with histological features |
| **H&E-like composite image** | Blurred and spatially overlapping (Fig 10b) | High spatial resolution, distinct spatial features(Fig 10a) |

Therefore, although PCA permits reconstructing the data with high accuracy, it does not generate readily interpretable information in each component spectrum. In contrast, NMF spectra are strictly non-negative. Therefore, spectral intensities carried by each component are always constructively superimposed during reconstruction. Consequently, if a given NMF component spectrum shows a significant ion, this peak will be clearly reflected in the reconstructed data. Hence, given a sufficiently low NMF reconstruction error, we can confidently state that the presence of an ion in an NMF component spectrum indicates that the lipid annotation corresponding to that mass-to-charge value was present in the tissue sample.

The results shown in Fig 7b establish that the PCA spatial and PCA spectral components exhibit distinct structures for low-order components, but the presence of distinct structures markedly decreases for higher-order components. This may be attributed to the fact that the PCA algorithm extracts components such that a maximum amount of variance in the data is captured in each successive component. Therefore, the higher-order PCA components will capture finer detail of the data such as pixel noise. In contrast, the NMF algorithm extracts components by optimizing the reconstruction error with a positivity constraint, thereby capturing differences inherent to the data in different components which together represent the whole data. It can be observed (Fig 7) that spatial structure is prominent only in the first few PCA components but quickly becomes blurred, whereas the spatial structure in NMF components is preserved for all components. In some cases, NMF spatial components also represent off-tissue or matrix components.

## Ranking NMF components

We have established a methodology for ranking the contribution of each NMF component to classification accuracy, with the top four components shown in Fig 8. The NMF spectra corresponding to the components fall into two general classes, those that indicate the predominant presence of single ions and those that reflect a more complex combination of ions and their relative abundances. For example, the significant peak in NMF component 2 aligns with the putative phospholipid SAPI which has been found in other work to correlate with inflammation [7]. Likewise, the significant peak in component 0 seems to correspond with the putative phospholipid PI 36:1, while the spectrum in component 10 reflects a complex combination of ions that is more indicative of something like a tissue fingerprint. These identification of compounds from the NMF peaks are from preliminary analysis and have not yet been validated experimentally. This is one of the future directions of our ongoing research project.

## Interpreting NMF spatial maps

As shown in Fig 10, by color-coding individual NMF spatial intensity maps, an equivalent to an H&E-stained image with better spatial specificity for different tissue types may be obtained. This is an unexpected result as there is no a-priori requirement for NMF components to capture anatomical information separately in its components. This ability to use NMF spatial intensity maps as a basis to generate 'H&E-stained like' images with higher spatial detail promises a link between the two techniques of MSI and H&E-stained imaging. There is good evidence here for an NMF-based H&E feature extraction tool that can automate the reading of well-defined tissue stains given enough training, and full automation of MSI-H&E coregistration based on automated regions of interest. We note that this study identifies useful roles for both the spectral and spatial components resulting from NMF feature extraction, with the spectral components providing interpretability into constituent ions and the spatial components enhancing the spatial interpretability of tissue structure and yielding high classification accuracy.

Attempts to align NMF spatial maps with H&E-stained images using approaches based on feature-matching were unsuccessful. This could be attributed to the graininess present in both NMF spatial maps and H&E-stained images, thereby precluding standard feature-based alignment techniques from extracting landmark features to generate a satisfactory alignment. We found that the homography transformation tuned by maximizing the enhanced correlation coefficient was successful in aligning NMF and H&E data.

## Limitations of NMF

Despite the demonstrated advantages of NMF feature extraction for MSI data, the computation itself has a relatively high cost. The requirement for high resolution in binning also increases the computational burden. At 0.05 Da bin size, an MSI dataset may occupy 20—30 GB of system memory. This leads to approximately 200 GB of data when the 8 datasets are stacked. While the PCA algorithm executes on this stacked data in under 10 minutes, approximately 24 hours were required for the convergence of the NMF algorithm. These bottlenecks may preclude the average user from using this technique on large datasets. However, the initial exploratory analysis could be achieved faster by using a larger bin width (0.1 Da). Indeed, similar results were obtained in preliminary analysis with bin widths of 1 Da. This issue could be resolved in future work by efficient computation with multiprocessing, better algorithms to compute NMF using GPU servers, and batch implementations of the NMF algorithm such that a large number of datasets could be processed with limited computational resources.

## Conclusion

This research was motivated by the observation that NMF provides effective feature extraction for MSI data, with individual NMF components exhibiting a strong correlation with underlying tissue structures and offering interpretability according to the *m/z* values of constituent molecular compounds. The novel contribution of this paper consists of three data processing methodologies that perform data augmentation to support the training and testing of classifiers, ranking of the most important features for classification, and image registration to support tissue-specific imaging. These methods are demonstrated on a novel MSI data cohort for a rodent model of chronic visceral pain.

The MSI data processing pipelines establish distinct roles for the spectral and spatial NMF components. The spectral components allow for interpretability in terms of the *m/z* values of constituent ions due to the nonnegativity constraint in NMF spectral decomposition. The spatial components not only enhance the contrast and spatial detail of tissue structures but also are distinctive enough to allow for high classification accuracy.

The overall advantage of these methodologies is spatial and spectral representations of MSI data that are directly interpretable, an observation that we have leveraged in conjunction with downstream analysis methods. We note that the PCA features exhibit similar or slightly higher performance than the NMF features for classification and reconstruction (Fig 4); the spectral and spatial interpretability of NMF features is a distinct advantage over PCA, allowing NMF components to be used directly for downstream data processing such as classification and data fusion without the need for an additional clustering step. The novel data augmentation technique allows data cohorts with a limited number of tissue samples to be used for training and testing unbiased classifiers. The novel feature ranking technique allows data analysis efforts to highlight components that are most discriminative between two experimental groups. This allows those components to be prioritized in subsequent investigations and analysis. The novel image registration technique allows NMF feature components with underlying correlation to tissue structure (as identified in H&E-stained images) to be identified and combined to

enhance specificity for different tissue types and anatomical structures. Image registration is required when establishing correlations between multiple experimental techniques that do not ensure registration at the scale of individual pixels.

The main disadvantage and limitation of these methodologies arises from their high computational cost. For the visceral pain data cohort with all 8 samples, execution of the NMF algorithm on a Dell Precision 5820 (Intel Core i9 10900X CPU with 20 cores) with 256 GB system memory required approximately 24 hours whereas execution of the PCA algorithm finished in about 10 minutes. The computational cost is further increased due to multiple samples of MSI data required in the data methodology presented here and limits the number of samples that can be processed.

Detailed descriptions of data processing pipelines are presented for 1) NMF feature extraction, 2) classification based on NMF features, and 3) image registration of NMF and H&E data. Three novel methodologies were developed for data augmentation, feature ranking, and image registration. The utility of these methods is demonstrated on a novel MSI data cohort for a rodent model of chronic visceral pain and supported by results including the successful and robust classification of naïve and co-morbid pain subjects as well as a meaningful interpretation of NMF features regarding tissue histology.

## Supporting information

**S1 Fig. NMF spatial maps for the data cohort.** Each row shows 20 NMF spatial maps for each of the 8 datasets. The first four rows represent the CPH data and the last four rows represent the naïve data.
(TIF)

**S2 Fig. PCA spatial maps for the data cohort.** Each row shows 20 PCA spatial maps for each of the 8 datasets. The first four rows represent the CPH data and the last four rows represent the naïve data.
(TIF)

**S3 Fig. Overlay of NMF spatial maps on the H&E images.** Overlay of 20 NMF spatial maps over the corresponding H&E image for each of the 8 datasets. The first four rows represent the CPH data and the last four rows represent the naïve data.
(TIF)

## Author Contributions

**Conceptualization:** Alison J. Scott, Richard J. Traub, Robert K. Ernst, Reza Ghodssi, Behtash Babadi, Pamela Ann Abshire.

**Data curation:** Alison J. Scott.

**Formal analysis:** Behtash Babadi.

**Funding acquisition:** Alison J. Scott, Richard J. Traub, Robert K. Ernst, Reza Ghodssi, Behtash Babadi, Pamela Ann Abshire.

**Investigation:** Kasun Pathirage, Aman Virmani, Alison J. Scott, Richard J. Traub, Robert K. Ernst, Reza Ghodssi, Behtash Babadi, Pamela Ann Abshire.

**Methodology:** Kasun Pathirage, Aman Virmani, Behtash Babadi, Pamela Ann Abshire.

**Project administration:** Robert K. Ernst, Reza Ghodssi, Behtash Babadi, Pamela Ann Abshire.

**Resources:** Alison J. Scott, Richard J. Traub, Robert K. Ernst, Reza Ghodssi, Behtash Babadi, Pamela Ann Abshire.

**Software:** Kasun Pathirage, Aman Virmani, Behtash Babadi.

**Supervision:** Richard J. Traub, Robert K. Ernst, Behtash Babadi, Pamela Ann Abshire.

**Validation:** Alison J. Scott.

**Visualization:** Kasun Pathirage, Aman Virmani.

**Writing – original draft:** Kasun Pathirage, Aman Virmani, Alison J. Scott, Richard J. Traub, Robert K. Ernst, Reza Ghodssi, Behtash Babadi, Pamela Ann Abshire.

**Writing – review & editing:** Kasun Pathirage, Aman Virmani, Alison J. Scott, Richard J. Traub, Robert K. Ernst, Reza Ghodssi, Behtash Babadi, Pamela Ann Abshire.

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
