## [Decision Letter · Decision Letter 0]

26 Jul 2023

PONE-D-23-15506Interpretable dimensionality reduction and classification of mass spectrometry imaging data in a visceral pain model via non-negative matrix factorizationPLOS ONE

Dear Dr. Abshire,

Thank you for submitting your manuscript to PLOS ONE. After careful consideration, we feel that it has merit but does not fully meet PLOS ONE’s publication criteria as it currently stands. Therefore, we invite you to submit a revised version of the manuscript that addresses the points raised during the review process.

We look forward to receiving your revised manuscript.

Kind regards,

Bardia Yousefi, Ph.D.

Academic Editor

PLOS ONE

Journal Requirements:

"This work was supported by a 2021 MPower Seed Grant from The University of Maryland Strategic Partnership: MPowering the State to authors PA, RE, RG, BB, AS, and RT. The funder website is 

https://mpower.maryland.edu/

We note that one or more of the authors is affiliated with the funding organization, indicating the funder may have had some role in the design, data collection, analysis or preparation of your manuscript for publication; in other words, the funder played an indirect role through the participation of the co-authors. If the funding organization did not play a role in the study design, data collection and analysis, decision to publish, or preparation of the manuscript and only provided financial support in the form of authors' salaries and/or research materials, please do the following:

(1) Review your statements relating to the author contributions, and ensure you have specifically and accurately indicated the role(s) that these authors had in your study. These amendments should be made in the online form.

(2) Confirm in your cover letter that you agree with the following statement, and we will change the online submission form on your behalf: 

4. We noted in your submission details that a portion of your manuscript may have been presented or published elsewhere:

"A version of this submission has been uploaded to bioRxiv.

" ext-link-type="uri" xlink:type="simple">https://www.biorxiv.org/content/10.1101/2023.04.24.538180v1"

Please clarify whether this publication was peer-reviewed and formally published. If this work was previously peer-reviewed and published, in the cover letter please provide the reason that this work does not constitute dual publication and should be included in the current manuscript.

7. Please ensure that you refer to Figure 3 in your text as, if accepted, production will need this reference to link the reader to the figure.

**Additional Editor Comments:**

This article has good merits, but needs a revision before it goes further. Thanks

Reviewers' comments:

Reviewer's Responses to Questions

**Comments to the Author**

1. Is the manuscript technically sound, and do the data support the conclusions?

Reviewer #1: Yes

Reviewer #2: Partly

2. Has the statistical analysis been performed appropriately and rigorously? 

Reviewer #1: Yes

Reviewer #2: No

3. Have the authors made all data underlying the findings in their manuscript fully available?

Reviewer #1: Yes

Reviewer #2: Yes

4. Is the manuscript presented in an intelligible fashion and written in standard English?

Reviewer #1: Yes

Reviewer #2: No

5. Review Comments to the Author

Reviewer #1: Interpretable dimensionality reduction and classification of mass spectrometry imaging data in a visceral pain model via non-negative matrix factorization

The authors tried to analyze the Mass spectrometry imaging (MSI) data using non-negative matrix factorization. NMF used to reduce dimensionality and encountering spatial components.

In my opinion, this manuscript has these good points:

- The subject is interesting and might absorb many readers in the field;

- Nice written and well presenting the idea;

- Suitable analytical representations;

Also, there are some suggestions that would increase the strength of the paper which is listed bellows;

- One of my major points about your article concerns the novelty of this article, authors should improve their novelty more highlighted. NMF is used for dimensionality reduction and spectroscopy data analysis (please google this to find more published contributions in the NMF for Hyperspectral, and spectroscopy), what is new in your article. Please specify your contributions.

- Why did the size of data shrink from 100K to 20? There should a gap statistic similar approach to justify this.

- Similarity to clustering is also needed to the be highlighted and how does this manifest itself into the analysis.

Thank you

Reviewer #2: In the presentation of the work, the article has a nice beginning; nonetheless, it has to be examined in order to help the reader comprehend “Interpretable dimensionality reduction and classification of mass spectrometry imaging data in a visceral pain model via non-negative matrix factorization." Despite the fact that I believe the work does not satisfy the requirements for publication in PLOS ONE and that there are some concerns that need to be addressed, I strongly propose a comprehensive review that will add value to the results that were acquired through discussion. The writers need to make some changes to the paper. On the other hand, I would like to provide the authors with the following remarks and suggestions:

1. The abstract does not communicate well; it must be revised.

2. In particular, it is not entirely evident how this publication contributes to the body of previous research when compared to other papers that have been published. Because of this, unable to propose that the current version be accepted.

3. In Background and related work section, the author can introduce more literature and analyse its shortcomings to highlight the advantages and innovations of this paper.

4. The results themselves need to be explained, which is why there must be a section or paragraph dedicated to the discussion along with an appropriate comparison table of the suggested work.

5. More specifically, only simulated results were presented, and there was no attempt made to verify the suggested work by practical means in the tabular form.

6. Add some recently proposed techniques (2020-2023) in the related work section of the manuscript.

7. There are too many spelling and grammar mistakes in the paper. It needs proper spelling and grammar checking.

8. Conclusion section should be extended by mentioning the advantages, disadvantages, and limitations of the study.

6. PLOS authors have the option to publish the peer review history of their article (what does this mean?). If published, this will include your full peer review and any attached files.

Reviewer #1: No

Reviewer #2: No

---

## [Author Response · Author response to Decision Letter 0]

30 Oct 2023

We thank the reviewers for their thoughtful feedback on our manuscript. We have now substantially revised our manuscript by including new analyses, clarifying our results, and improving the clarity and rigor of our expositions. In summary, the major changes include:

1) Adding new analyses to justify the choice of the reduced dimensions (Fig. 4b) and extending the comparison of our proposed approach with existing methods (Fig. 10b and Table 1); 

2) Substantially revising the abstract to emphasize the novel contributions of the paper and to be more accessible for a general audience;

3) Clarifying our main contributions in the context of existing recent work, in both the background and discussion sections, and highlighting the advantages, disadvantages, and limitations of our approach;

4) Substantially revising and enhancing the conclusion section; and

5) Improving the rigor and clarity of our presentation by significantly revising the text throughout the manuscript.

In what follows, we respond to the comments of the reviewers in a point-by-point fashion.

We would like to thank the reviewing editor and the anonymous reviewers for their careful critique of our work and for their constructive and thorough feedback.

Reviewer #1:

Interpretable dimensionality reduction and classification of mass spectrometry imaging data in a visceral pain model via non-negative matrix factorization

The authors tried to analyze the Mass spectrometry imaging (MSI) data using non-negative matrix factorization. NMF used to reduce dimensionality and encountering spatial components.

In my opinion, this manuscript has these good points:

- The subject is interesting and might absorb many readers in the field;

- Nice written and well presenting the idea;

- Suitable analytical representations;

Response: We thank the reviewer for summarizing the strengths of our contributions.

Reviewer #1:

Also, there are some suggestions that would increase the strength of the paper which is listed bellows;

- One of my major points about your article concerns the novelty of this article, authors should improve their novelty more highlighted. NMF is used for dimensionality reduction and spectroscopy data analysis (please google this to find more published contributions in the NMF for Hyperspectral, and spectroscopy), what is new in your article. Please specify your contributions.

Response: We thank the reviewer for raising this point on clarifying the novelty of our work. We have now revised our abstract to clarify the main contributions of the work as being three novel MSI data analysis techniques that leverage the spatial and spectral interpretability of NMF compressed MSI data (pp. 1 - 2).

We also added a section highlighting the connection to existing work in MSI data analysis to highlight the novel aspects of this work (pp. 5 - 6).

Reviewer #1:

- Why did the size of data shrink from 100K to 20? There should a gap statistic similar approach to justify this.

Response: 

We are not performing a traditional clustering, so gap statistics are not readily available or well defined for this approach. Instead we set a target error rate of 20% in reconstruction accuracy and used this criterion to select the number of components to retain in subsequent analysis. We have now revised Fig 4 to clearly show the selection criterion for the reduced dimension (for both PCA and NMF) (pp. 13)

Reviewer #1:

- Similarity to clustering is also needed to the be highlighted and how does this manifest itself into the analysis.

Response: One of the key findings in this work is that NMF inherently produces components with spatially distinct structure obviating the need for an additional explicit clustering step. We have now added this explanation to the section on review of existing MSI data processing methods (pp. 5 - 6).

Reviewer #2:

In the presentation of the work, the article has a nice beginning; nonetheless, it has to be examined in order to help the reader comprehend “Interpretable dimensionality reduction and classification of mass spectrometry imaging data in a visceral pain model via non-negative matrix factorization." Despite the fact that I believe the work does not satisfy the requirements for publication in PLOS ONE and that there are some concerns that need to be addressed, I strongly propose a comprehensive review that will add value to the results that were acquired through discussion. The writers need to make some changes to the paper. On the other hand, I would like to provide the authors with the following remarks and suggestions:

Response: We thank the reviewer for their careful critique of our work and for providing a number of insightful suggestions to improve it.

Reviewer #2:

1. The abstract does not communicate well; it must be revised.

Response: We have now revised our abstract to clarify the main contributions of the work as being three novel MSI data analysis techniques that leverage the spatial and spectral interpretability of NMF compressed MSI data (pp. 1 - 2). 

Reviewer #2:

2. In particular, it is not entirely evident how this publication contributes to the body of previous research when compared to other papers that have been published. Because of this, unable to propose that the current version be accepted.

Response: As noted above, we have now revised the abstract to better communicate the novelty of our work (pp. 1 - 2).

We have also added a more thorough literature review, specifically highlighting the novelty of our work in terms of NMF as it applies to downstream MSI data processing (pp. 5 - 6).

Reviewer #2:

3. In Background and related work section, the author can introduce more literature and analyze its shortcomings to highlight the advantages and innovations of this paper.

Response: We have now added a more thorough literature review, specifically highlighting the novelty of our work on NMF data compression as it applies to downstream MSI data processing. We have also emphasized how our work relates to existing approaches (pp. 5 - 6).

Reviewer #2:

4. The results themselves need to be explained, which is why there must be a section or paragraph dedicated to the discussion along with an appropriate comparison table of the suggested work.

Response: We have added subheadings in the discussion section to highlight the material related to results for distinct topics in our paper and improve readability. (pp. 22- 25)

Since the novelty of our work lies in data processing methodologies (i.e., how to integrate NMF into data processing pipelines for downstream analyses), it is hard to perform a quantitative comparison with other methods. 

To establish a benchmark for how NMF and PCA perform on the novel dataset used in this work, we have added a table illustrating the comparison of PCA and NMF results on this novel dataset (pp. 22).

We have also modified Fig 10 to show how the methodologies described in this paper, when applied to NMF compressed MSI data, compare against MSI data compressed with PCA. (pp. 21)

Reviewer #2:

5. More specifically, only simulated results were presented, and there was no attempt made to verify the suggested work by practical means in the tabular form.

Response: All of the results in this paper are based on a novel MSI dataset obtained from an animal model of comorbid visceral pain hypersensitivity, which is described in the Animal Model, Tissue Preparation, Mass spectrometry imaging and staining, and Datasets sections (pp. 7-9), We have reworded the abstract to emphasize the novelty of this dataset that we use for validating the methods (pp. 1 - 2). As noted above, we have also added a table comparing PCA and NMF approaches as applied to this novel dataset (pp. 22).

Reviewer #2:

6. Add some recently proposed techniques (2020-2023) in the related work section of the manuscript.

Response: We have added a section on MSI data processing highlighting recent contributions and their connection to this work. We have updated the bibliography with the respective literature (references 21 - 24) (pp. 5 - 6).

Reviewer #2:

7. There are too many spelling and grammar mistakes in the paper. It needs proper spelling and grammar checking.

Response: We have thoroughly proofread the manuscript for spelling and grammatical mistakes and have resolved all issues that we found.

Reviewer #2:

8. Conclusion section should be extended by mentioning the advantages, disadvantages, and limitations of the study.

Response: We have significantly expanded the conclusion, describing in detail the advantages, disadvantages and limitations of our study (pp. 25 - 26).

---

## [Decision Letter · Decision Letter 1]

29 Feb 2024

Interpretable dimensionality reduction and classification of mass spectrometry imaging data in a visceral pain model via non-negative matrix factorization

PONE-D-23-15506R1

Dear Dr. Abshire,

We’re pleased to inform you that your manuscript has been judged scientifically suitable for publication and will be formally accepted for publication once it meets all outstanding technical requirements.

Kind regards,

Bardia Yousefi, Ph.D.

Academic Editor

PLOS ONE

Additional Editor Comments (optional):

Authors responded well to the comments received. I recommend accepting this manuscript

Congratulations

Reviewers' comments:

Reviewer's Responses to Questions

**Comments to the Author**

1. If the authors have adequately addressed your comments raised in a previous round of review and you feel that this manuscript is now acceptable for publication, you may indicate that here to bypass the “Comments to the Author” section, enter your conflict of interest statement in the “Confidential to Editor” section, and submit your "Accept" recommendation.

Reviewer #1: All comments have been addressed

2. Is the manuscript technically sound, and do the data support the conclusions?

Reviewer #1: Yes

3. Has the statistical analysis been performed appropriately and rigorously? 

Reviewer #1: Yes

4. Have the authors made all data underlying the findings in their manuscript fully available?

Reviewer #1: Yes

5. Is the manuscript presented in an intelligible fashion and written in standard English?

Reviewer #1: Yes

6. Review Comments to the Author

Reviewer #1: Authors responded well to my comments.

Particularly respond to the novelty was sufficient.

I don't have any comments.

7. PLOS authors have the option to publish the peer review history of their article (what does this mean?). If published, this will include your full peer review and any attached files.

Reviewer #1: No

---

## [Editor Report · Acceptance letter]

21 Mar 2024

PONE-D-23-15506R1 

PLOS ONE

Dear Dr. Abshire, 

I'm pleased to inform you that your manuscript has been deemed suitable for publication in PLOS ONE. Congratulations! Your manuscript is now being handed over to our production team.

Kind regards, 

on behalf of

Dr. Bardia Yousefi 

Academic Editor

PLOS ONE